# Identification and Characterization of *Nigrospora* Species and a Novel Species, *Nigrospora anhuiensis*, Causing Black Leaf Spot on Rice and Wild Rice in the Anhui Province of China

**DOI:** 10.3390/jof10020156

**Published:** 2024-02-16

**Authors:** Yang Liu, Jiahao An, Asma Safdar, Yang Shen, Yang Sun, Wenhui Shu, Xiaojuan Tan, Bo Zhu, Jiaxin Xiao, Jan Schirawski, Feng He, Guoping Zhu

**Affiliations:** 1College of Life Sciences, Anhui Normal University, Wuhu 241000, China; 17805591959@163.com (Y.L.); 15883917426@163.com (J.A.); m17356123402@163.com (Y.S.); sy2019@ahnu.edu.cn (Y.S.); 18297471558@163.com (W.S.); xjtan@ahnu.edu.cn (X.T.); zhubo_ybdx119@163.com (B.Z.); xjx0930@163.com (J.X.); 2Department of Plant Pathology, College of Agriculture, University of Sargodha, Sargodha 40100, Pakistan; 3Department of Genetics, Matthias Schleiden Institute, Friedrich Schiller University Jena, 07743 Jena, Germany; jan.schirawski@uni-jena.de

**Keywords:** fungal pathogens, phylogenetic tree, rice, *Nigrospora anhuiensis*, black leaf spot disease, fungicide, *Nigrospora oryzae*

## Abstract

Rice production in the Anhui province is threatened by fungal diseases. We obtained twenty-five fungal isolates from rice and wild rice leaves showing leaf spot disease collected along the Yangtze River. A phylogenetic analysis based on internal transcribed spacer (ITS), translation elongation factor 1 alpha (*TEF1-α*), and beta tubulin (*TUB2*) sequences revealed one isolate (SS-2-JB-1B) grouped with *Nigrospora sphaerica*, one (QY) with *Nigrospora chinensis*, twenty-two with *Nigrospora oryzae*, and one isolate (QY-2) grouped in its own clade, which are related to but clearly different from *N. oryzae*. Nineteen tested isolates, including sixteen strains from the *N. oryzae* clade and the three isolates of the other three clades, caused disease on detached rice leaves. The three isolates that did not belong to *N. oryzae* were also able to cause disease in rice seedlings, suggesting that they were rice pathogens. Isolate QY-2 differed from the other isolates in terms of colony morphology, cell size, and susceptibility to fungicides, indicating that this isolate represents a new species that we named *Nigrospora anhuiensis*. Our analysis showed that *N. sphaerica*, *N. chinensis*, and the new species, *N. anhuiensis*, can cause rice leaf spot disease in the field. This research provides new knowledge for understanding rice leaf spot disease.

## 1. Introduction

*Nigrospora* species are ascomycetes and include both endophytic and parasitic fungi that are becoming famous [1,2,3,4]. This group of fungi can infect a wide range of plants, thereby causing great economic losses in crop production [2,3,5,6,7,8,9]. *Nigrospora* species are known to infect *Dioscorea opposita* [10], *Oryza satiwa* [11], *Chrysanthemum moratorium* [3], *Passiflora edulis* Sims [12,13], *Arachis hypogaea* [14], *Nicotiana tabacum* [15], and *Photinia serrulata* [16] in China and lead to leaf spot or blight symptoms. They seem to have a broad host range. The known rice pathogen *Nigrospora oryzae*, for example, can also infect yam, wild rice, poplar, kiwifruit, red elephant grass, cotton, and pear [3,8,10,11,17,18,19,20]. In light of the global temperature increase, *Nigrospora* species will likely cause numerous plant diseases. Thus, the threat caused by *Nigrospora* species should be drawing our attention, and more knowledge about these pathogens should be gained. 

Initially, the characterization of *Nigrospora* species was based on morphological features, and species identification was performed via a sequence comparison of the internal transcribed spacer (ITS) region [2,5,8,21]. However, among the members of this species, there are very few morphological differences that are limited to differences in colony color, spore size, and spore color [21]. Recently, multi-locus sequence analysis was used for the identification of fungal species. According to an analysis based on a multi-locus phylogenetic tree that was constructed using the combined ITS, *TEF1-α*, and *TUB2* sequences, *Nigrospora* species were divided into sixteen groups, including the famous rice pathogen *N. oryzae*, and two unclear groups [2,5,6,10,21]. This analysis also led to the identification of many new species [9,22]. Therefore, species identification based on the multi-locus sequence analysis is more reliable and will be especially beneficial for the characterization of *Nigrospora* species lacking special morphological features.

*N. oryzae*, *Nigrospora panici*, and *Nigrospora chinensis* are all known to inhabit rice leaves as endophytes [23]. However, *N. oryzae* is also known as a rice pathogen that causes yellow-to-brown, oval, or circular lesions with large yellow halos on rice leaves [11] and is universally distributed on various plants or dead plant residues in the field. Initially, many of the *Nigrospora* species were classified as *N. oryzae* because of the high similarity of their ITS sequences [23]. Therefore, many identified isolates of *N. oryzae* are probably not completely accurately classified. The correct species identification of rice pathogens is a prerequisite for the successful development of novel rice disease-control strategies. 

The provinces along the Yangtze River are important rice production regions in China. Especially in the Anhui province, where the weather is suitable for fungal growth and transmission, there is an abundant fungal biodiversity. Recently, *Nigrospora* species were discovered to cause leaf spot disease on rice [11]. However, there is little known about this species in the cities along the Yangtze River in the Anhui province. In this study, we investigated rice and wild rice leaf spot diseases in six cities of the Anhui province. Following the collection of diseased rice samples with oval or circular lesions, a molecular analysis was performed. This revealed the discovery of three novel rice pathogens in addition to the known *N. oryzae*: *Nigrospora sphaerica*, *N. chinensis*, and a new species, *Nigrospora anhuiensis*.

## 2. Materials and Methods

### 2.1. Sample Collection of Diseased Leaves and Pathogen Isolation

For sample collection, we screened at least five rice fields each (each field > 1000 acres) at 17 sites near different cities in the Anhui province along the Yangtze River during the rice production season from June to October 2021. Instances of rice leaf spot disease were recorded, and samples of diseased rice leaves were collected. We also screened and sampled the wild rice and grasses near the rice fields. To isolate fungal plant pathogens, collected detached leaves were surface-sterilized by rinsing in 75% ethanol for 30 s and 0.5% sodium hypochlorite for 5 min, followed by three rinses in sterile distilled water, their drying, cutting in 0.5 cm strips, and placement on potato dextrose agar (PDA) containing kanamycin (50 mg/L). Individual isolates were sub-cultured three times on kanamycin-containing PDA. Fungal morphology was recorded by photography after two and four days of incubation at 25 °C.

### 2.2. Phylogenetic Analysis of Fungal Isolates 

Mycelia of fungal isolates were scraped off from PDA plates that were incubated for 2 days at 25 °C. The material was collected and used for genomic DNA isolation [24]. The ITS, *TEF1-α*, and *TUB2* sequences of the isolates were amplified using primer pairs according to a previous study [23]. PCR products were sent for sequencing by the Sangon Company (Shanghai). The newly generated sequences were complemented with the ITS, *TEF1-α*, and *TUB2* sequences from different *Nigrospora* species available in the NCBI nucleotide database [21], and accession numbers for each gene are presented in Table 1. Nonsense sequences were removed from 5′ and 3′ flanks prior to sequence alignment using MEGA7.0 [25]. The phylogenetic tree was generated using the neighbor-joining method with 1000 bootstrap replicates with the MEGA7.0 software [25]. ITS, *TEF1-α*, and *TUB2* sequences of the new *N. anhuiensis* isolates were submitted to the NCBI database (Table 1).

### 2.3. Virulence Testing of Fungal Isolates 

For testing virulence of fungal isolates on rice, the fungal isolates were first cultured on PDA for two days. Agar plugs from fungal colony edges were excised from the plates and positioned on detached leaves of six-week-old rice seedlings or on six-week-old rice seedlings grown under natural conditions in the summer, and the sterile PDA plugs were used as control. The inoculated detached rice leaves were incubated at 25 °C for two days, at which lesions were observed and lesion sizes were measured. Each strain was used to inoculate at least three leaves. Lesion sizes occurring on the inoculated seedlings grown under natural conditions were observed one week post-inoculation. Three seedlings were inoculated per fungal isolate. Data for the lesion length were analyzed using one-way ANOVA through SPSS 25 (SPSS version 25; IBM, Chicago, IL, USA), and the QY-2 isolate was used as the reference for ANOVA analysis.

### 2.4. Analysis of Morphological Features of the Isolates

To observe morphologic similarities and differences of the isolate QY-2 with *N. oryzae*, *N. sphaerica* SS-2JB-1B, or *N. chinensis* QY, all isolates were freshly grown on potato dextrose agar (PDA) and colony features were observed after incubation for two and four days. To observe the features of individual hyphae, we cultured the strains for 24 h on PDA that was spread on sterile microscopic slides and imaged the hyphae using bright light illumination of a fluorescence inverted microscope (Leica, Wetzlar, Germany). We induced conidia formation of the five isolates by cultivation on carrot agar (200 g/L fresh carrot soup, 15 g/L agar powder) at 25 °C for seven days. Conidia were collected from the agar plates and analyzed using microscopy (Leica, Wetzlar, Germany). Of each species, ten conidia were measured in length, and the data were analyzed using Prism GraphPad 8.0 [26].

### 2.5. Sensitivity Testing of N. anhuiensis and N. oryzae to Eight Different Fungicides

To test sensitivity of *N. anhuiensis* QY-2 to commercial fungicides, we selected the eight fungicides, chlorothalonil, carbendazim, thifluzamide, flusilazole, thiophanate–methyl, mancozeb, difenoconazole, and prochloraz (Meilinxuehai, Nanjing, China), that are commonly applied in the field for fungal disease control. The QY-2 strain was cultured on the plates with different concentrations (0, 2.5, 5, and 10 µg/mL) of each fungicide, and *N. oryzae* isolate SS2-1A was used as control. Colony diameters for each plate were measured and photographed at two days of incubation at 25 °C. For each concentration, three technical replicates were prepared. Colony diameters were calculated using the formula Y = (D_n_ − D_0_) × 100/D_0_, where D_0_ indicates the colony diameter on PDA, and D_n_ denotes the colony diameter on fungicide containing PDA and analyzed using one-way ANOVA through SPSS 25 (SPSS version 25; IBM, Chicago, IL, USA). 

## 3. Results

### 3.1. Nigrospora Species Are Associated with Leaf Spot Disease in the Anhui Province, China

The Anhui province is one of the major rice-producing areas in China, located in the eastern part of the Yangtze River basin. To investigate incidences of rice disease in the Anhui province, we screened a total area of more than 17,000 acres of rice fields and neighboring wild rice and grass plants. Incidences of rice disease in the form of circular or oval lesions on rice leaves were widely observed in Qingyang county, Nanling county, and Guichi district, and a few incidences were found in Tongling City and Xuancheng City. Most of the samples were found in private rice fields, where the disease incidence was below 0.1%. In fields managed by companies or institutes, only very few diseased leaves were found. The diseased leaves showed leaf blight and brown-to-black circular, oval, or fusiform lesions, which had a halo of yellow or brown (Figure 1). We isolated and purified leaf-colonizing fungi from diseased rice or wild rice leaves collected at various sites in the Anhui province, China, resulting in a total of 25 novel isolates (Table 2).

### 3.2. Isolate QY-2 Is a Novel Nigrospora Species Based on Multi-Locus Phylogenetic Analysis

To allow species identification of the novel isolates, we amplified and sequenced their ITS, *TEF1-α,* and *TUB2* genomic regions and used the obtained sequences for multi-locus phylogenetic tree construction after having added available reference sequences of different *Nigrospora* species. Most (22 of 25) isolates clustered with known *N. oryzae* isolates and are, thus, identified as *N. oryzae* isolates (Figure 2; black ovals). One isolate (SS-2JB-1B) was grouped with *N. sphaerica* isolates (Figure 2; green triangle), one (QY) with *N. chinensis* isolates (Figure 2; blue triangle), and one isolate (QY-2) was located on a separate branch with a high statistical score that was related to, but clearly different than, *N. oryzae* (Figure 2; red triangle). Thus, rice disease symptoms in the Anhui province could be associated with fungal isolates of *N. oryzae*, *N. sphaerica*, *N. chinensis*, and a novel *Nigrospora* spp. that is related to *N. oryzae*. We named the novel *Nigrospora* species that is represented by isolate QY-2 *N. anhuiensis*.

### 3.3. The New N. anhuiensis Isolate QY-2 Has Similar Morphologic Features as Other Nigrospora Species

We aimed to observe whether the new *N. anhuiensis* isolate QY-2 could be morphologically differentiated from the well-characterized *N. oryzae* rice pathogen. For comparison, we included in our analysis the less closely related isolates *N. sphaerica* SS-2JB-1B and *N. chinensis* QY. *N. anhuiensis* QY-2 formed a white colony with few aerial mycelia at 2 days of incubation on nutrient agar, which is similar to that of *N. oryzae* isolates (Figure 3A). At four days of incubation, the *N. anhuiensis* QY-2 colony center changed its color to dark green, and the colony formed massive white aerial hyphae (Figure 3A). In contrast, the colony color of the two *N. oryzae* isolates was palm green or brown, and the massive aerial mycelia formed were grey (Figure 3A). *N. sphaerica* SS-2JB-1B and *N. chinensis* QY isolates also produced grey aerial mycelia, but the morphology of the aerial mycelia was different from that of the other three isolates (Figure 3A). All investigated isolates formed smooth, branched, and septate hyphae (Figure 3B). Conidia formation of all isolates could be induced on carrot medium. Conidia of all investigated isolates had a similar globose or oval, black, smooth, and aseptate morphology, while several grey and transparent conidia were also found in QY-2, SS2-1A, and QY isolate (Figure 3C). The sizes of the conidia varied slightly among the isolates. For *N. anhuiensis* QY-2, conidia diameter was, on average, 12.75 µm but varied by 3 µm between the smallest and the largest measured conidium. Conidia of the *N. oryzae* isolates were, on average, 12.90 and 13.40 µm but varied only by 1 µm between the smallest and the largest measured conidium (Figure 4). The conidia of *N. sphaerica* SS-2JB-1B and *N. chinensis* QY were 13.1 and 13.3 µm on average, with a 1 µm range between the smallest and the largest measured conidium (Figure 4). Based on morphological criteria, *N. anhuiensis* can best be differentiated from *N. oryzae*, *N. sphaerica*, and *N. chinensis* by its dark green colony center color at four days of incubation on nutrient agar. However, we recommend species identification by multigene sequencing. 

### 3.4. The Identified N. oryzae, N. sphaerica, N. chinensis, and N. anhuiensis Isolates Are Rice Pathogens 

To check whether the isolated isolates were rice pathogens, we performed a virulence test on detached rice leaves. Because some of the isolates were collected in the same field and also are *N. oryzae* isolates, we selected 16 isolates, including most *N. oryzae* isolates, for virulence testing. A total of 19 of 25 isolates were able to induce black spot lesions on detached rice leaves (Figure 5). This included two isolates that were isolated from wild rice and the identified *N. sphaerica*, *N. chinensis*, and *N. anhuiensis* isolates (Figure 5). 

To test whether the isolates were able to cause disease on plants cultivated in the field, we performed a virulence test on rice and wild rice seedlings that were inoculated with either *N. oryzae* isolates SS2-1A, SS-2-JB-2A, *N. sphaerica* isolate SS-2JB-1B, *N. chinensis* isolate QY, or *N. anhuiensis* isolate QY-2. The inoculated rice seedlings were incubated in natural conditions in the summer. All isolates induced lesions on healthy rice plants. Lesion length was largest with *N. anhuiensis* isolate QY-2, and *N. sphaerica* isolate SS-2JB-1B induced the smallest lesions (Figure 6). The lesions caused by *N. anhuiensis* isolate QY-2 were brown to black and irregular, while the *N. sphaerica* isolate SS-2JB-1B and the *N. chinensis* isolate QY caused fusiform lesions with yellow verges (Figure 6). These symptoms were similar to the symptoms in the field. We re-isolated the fungi from the lesions, and the colony features were the same as those of the isolates used for infection. In summary, we conclude that the *N. sphaerica*, *N. chinensis*, and *N. anhuiensis* isolates are the agents that cause rice or wild rice leaf spot disease. 

### 3.5. Sensitivity of N. anhuiensis Isolate QY-2 and N. oryzae Isolate SS2-1A to Various Fungicides

To investigate whether the newly identified rice pathogen *N. anhuiensis* can be killed by commercial fungicides, we tested the eight fungicides chlorothalonil, carbendazim, thifluzamide, flusilazole, thiophanate–methyl, mancozeb, difenoconazole, and prochloraz that were already registered in China for their capacity to inhibit growth of *N. anhuiensis* isolate QY-2 and *N. oryzae* isolate SS2-1A on nutrient agar plates. We observed that all eight tested fungicides could inhibit the mycelial growth of both isolates in a concentration-dependent manner (Figure 7). For most fungicides, *N. anhuiensis* isolate QY-2 was more sensitive than *N. oryzae* isolate SS2-1A (Figure 7). Carbendazim was the most effective fungicide against *N. anhuiensis* isolate QY-2, with a concentration of 5 µg/mL suppressing its growth almost completely (Figure 7). *N. anhuiensis* isolate QY-2 could still grow on plates containing 10 µg/mL of thifluzamide, flusilazole, mancozeb, and difenoconazole, which suggests that these fungicides might be less effective in eradicating infection with *N. anhuiensis* in the field (Figure 7). At lower fungicide concentrations, the colony diameter of *N. oryzae* isolate SS2-1A was dramatically bigger than that of *N. anhuiensis* isolate QY-2 for the fungicides difenoconazole, chlorothalonil, thiophanate–methyl, and thifluzamide, suggesting that *N. oryzae* isolate SS2-1A was more resistant to these fungicides (Figure 7). Taken together, these data imply that the *N. anhuiensis* isolate QY-2 is sensitive to all tested fungicides. However, the growth of the isolate in vitro can be effectively inhibited by carbendazim, which might be the best choice for controlling *N. anhuiensis* rice spot disease spread in the field.

## 4. Discussion

This study was conducted to identify the diversity and number of *Nigrospora* species prevailing in infected rice fields in China. Leaf spot symptoms were observed in rice fields that were similar to those observed in other studies, i.e., brown-to-black fusiform-shaped lesions on leaves, branches, and pedicles [11]. In contrast to previous observations, the lesions observed in this study were surrounded by a yellow halo. *Nigrospora* species were initially considered as endophytic fungi [27,28] but were recently considered as the causal agent of leaf spot disease on rice plants [11]. Since then, many *Nigrospora* species have been identified as potential pathogens of new plant diseases on various host plants in different regions. This indicates that the *Nigrospora* species may threaten crop production. We should pay more attention to this species. 

Recently, fungal identifications were based on morphology and molecular analysis, including the similarity of ITS or multi-genes [29]. However, the identification of the isolated isolates solely based on morphological characters does not seem to be reliable. Different fungi are separated using various multi-genes [2,29,30]. In *Nigrospora species*, the isolates were separated using the gene combination of ITS, *TEF1-α*, and *TUB2* [3,4,11]. In this study, we also constructed ITS, *TEF1-α*, and *TUB2* multigene-based identification and phylogeny analysis, which resulted in reliable species identification. A total of twenty-five isolates of *Nigrospora* species were classified into *N. oryzae*, *N. sphaerica*, *N*. *chinensis*, and a new species that we named *N. anhuiensis*. This shows that multigene phylogenetic analysis is a reliable and powerful method to differentiate morphologically difficult-to-identify fungal isolates. However, even this analysis method provides challenges for species identification, such as the high similarity of gene sequences, which makes the clear identification of some isolates more difficult. The identification method should, thus, receive further attention in the future.

Among the recovered species, we most often found *N. oryzae*, which is a well-known rice pathogen causing leaf spot disease [11]. However, we also isolated three additional species that were not known to be associated with leaf spot disease on rice: *N. sphaerica*, *N. chinensis*, and the newly identified species *N. anhuiensis*. We experimentally showed that all isolates were able to infect rice plants and cause spot lesions on the rice leaves and stalks. These results imply that rice leaf spot disease is not only caused by *N. oryzae* but also by *N. sphaerica*, *N*. *chinensis*, and *N. anhuiensis*. Furthermore, wild rice that produces wild rice shoots was widely cultivated around the rice fields in the Anhui province of China. The leaf spot disease caused by *N. oryzae* also occurred on wild rice leaves and grass, which may be another threat to rice disease control in Anhui province. These discoveries not only enlarge the known diversity of *Nigrospora* species in the Anhui province but also provide new clues to improve rice field management. 

For new fungal species characterizations, combined morphological and molecular analyses are generally performed. However, there are few typical morphological characteristics for all the *Nigrospora* species. In our study, we discovered a novel isolate, QY-2, that has a smaller minimum conidia size, while the conidia size of *N. oryzae* isolates is bigger [11,17]. The conidia size is a special feature for fungal species identification. However, there are a few differences between the *Nigrospora* isolates, and the conidia size of the two *N. oryzae* isolates SS2-1A and SS-2-JB-2A are also different. Therefore, the morphological features are not reliable. The phylogenetic tree showed that QY-2 was classified into a single clade, suggesting it should belong to a new species named *N. anhuiensis* QY-2. The clade of QY-2 is closely related to *N. oryzae*, and the disease symptoms caused by the isolate QY-2 are also similar to those of *N. oryzae* following Koch’s postulates. Furthermore, our analysis showed that the isolate QY-2 is more sensitive to most tested fungicides than *N. oryzae* SS2-1A, suggesting that it can be easily controlled by general fungicides in the rice field. Therefore, it will likely not cause unpredictable yield loss when rice field management is carried out promptly.

### New Species Description

*Nigrospora anhuiensis* Y. Liu & F. He, sp. nov.—Fungal Names|FN571838; Figure 2 and Figure 3. Named after the Anhui province of China, where this species was first collected. The isolate infects rice leaves and straw, causing leaf spot symptoms with a yellow halo. Hyphae hyaline, branched, septate. Conidia solitary, subglobose, black, shiny, smooth, aseptate, ellipsoidal, 11.715 − 13.998 × 9.62 − 11.418 μm (av. = 13.136 ± 0.815 × 10.574 ± 0.644). Colonies grow quickly on PDA and are initially white, followed by dark green with aging, reaching 6 cm in 2 d at 25 °C. On the carrot medium, the conidia will be produced in a week. Isolate of *Nigrospora anhuiensis* are classified in a distinct clade on a phylogenetic tree with a high support value (Figure 2).

## 5. Conclusions

In this study, we investigated the occurrence of the rice black leaf spot disease in the Anhui province of China and found that not only *N. oryzae* but also *N. sphaerica*, *N. chinensis*, or *N. anhuiensis* can cause rice black leaf spot disease. *N. anhuiensis* is a novel *Nigrospora* species based on a multi-locus phylogenetic tree and morphological analysis. The study provides new knowledge about rice leaf spot disease and suggests potential chemical agents to control rice black leaf spot disease in the field.

## Figures and Tables

**Figure 1 jof-10-00156-f001:**
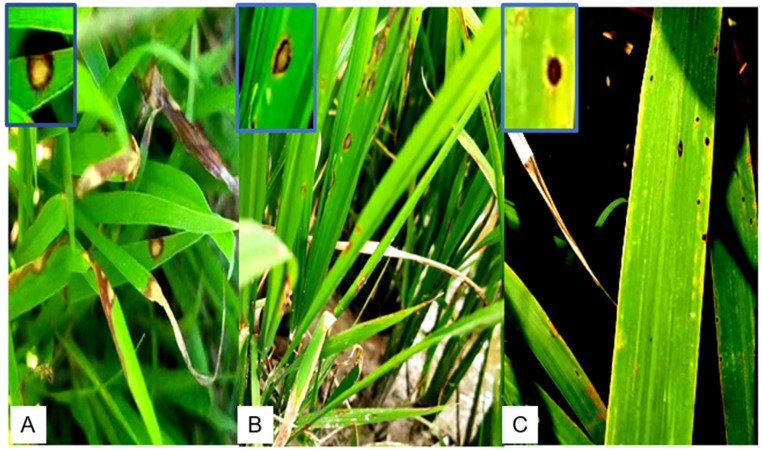
Leaf spot symptoms caused by *Nigrospora* species in the field. (**A**) Grass leaf spot disease in the rice field. (**B**) Rice leaf spot symptoms. (**C**) Wild rice leaf spot symptoms.

**Figure 2 jof-10-00156-f002:**
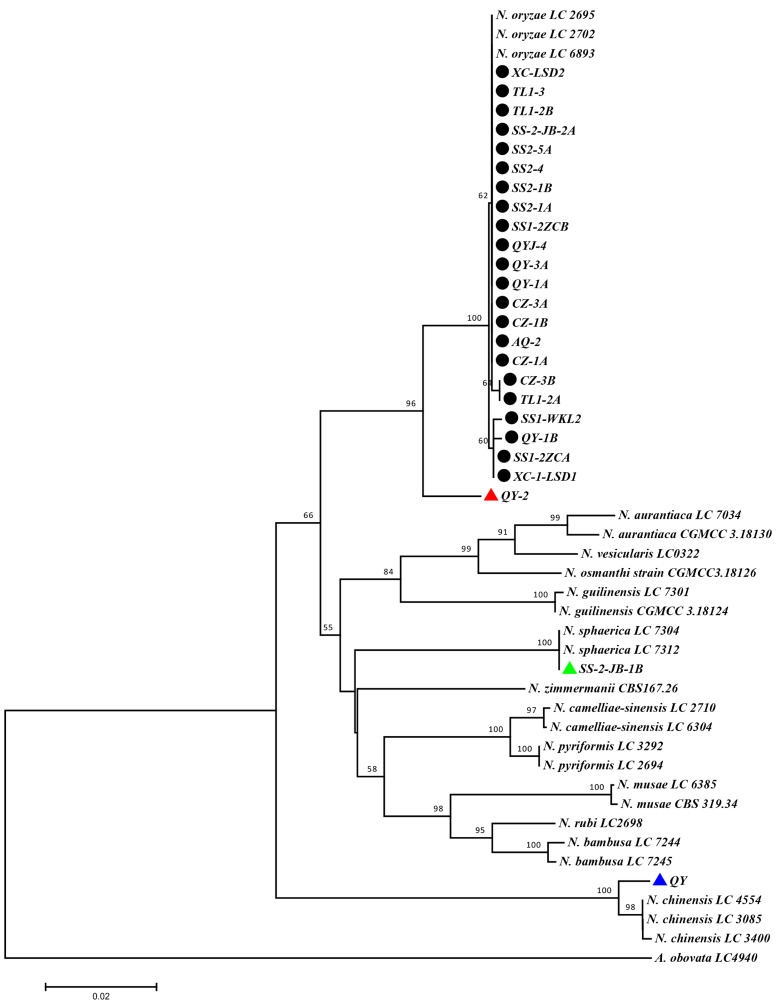
Phylogenetic tree based on the combined sequences of ITS, *TEF1a*, and *TUB2*. The ITS, *TEF1-α*, and *TUB2* sequences were obtained from the NCBI database or by sequencing. The combined sequences were aligned using ClustalW2. The phylogenetic tree was generated through the neighbor-joining method with 1000 bootstrap replicates using MEGA 7.0. The isolates of this study are marked on the tree (black oval, *N. oryzae* isolates; red triangle, *N. anhuiensis* isolate; green triangle, *N. sphaerica* isolate; blue triangle, *N. chinensis* isolate). Size marker shows the evolutionary distances.

**Figure 3 jof-10-00156-f003:**
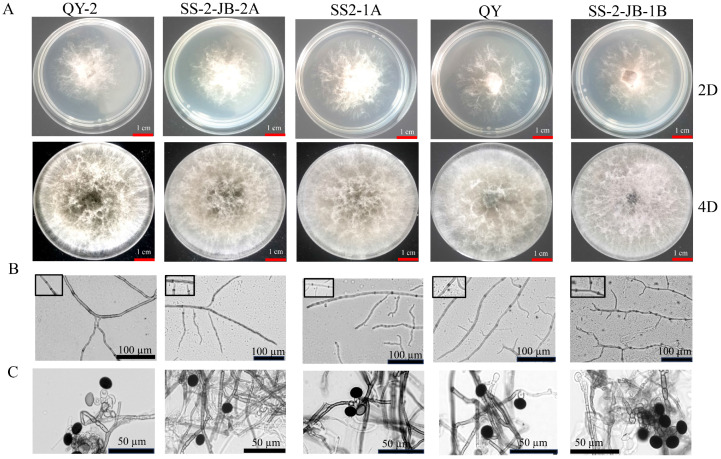
Morphological features of five *Nigrospora* isolates. (**A**) The colony features of five typical isolates. The isolates were cultured on PDA medium, and pictures were taken at 2 days (2D) and 4 days (4D) post-inoculation. (**B**) Mycelial features. The isolates were cultured on glass slides for 24 h. Pictures were taken using a microscopy imaging system (Leica, Wetzlar, Germany). (**C**) Conidia features. The conidia were induced on carrot medium for one week and imaged using a microscopy imaging system (Leica, Wetzlar, Germany). The scale bar is shown in the photos.

**Figure 4 jof-10-00156-f004:**
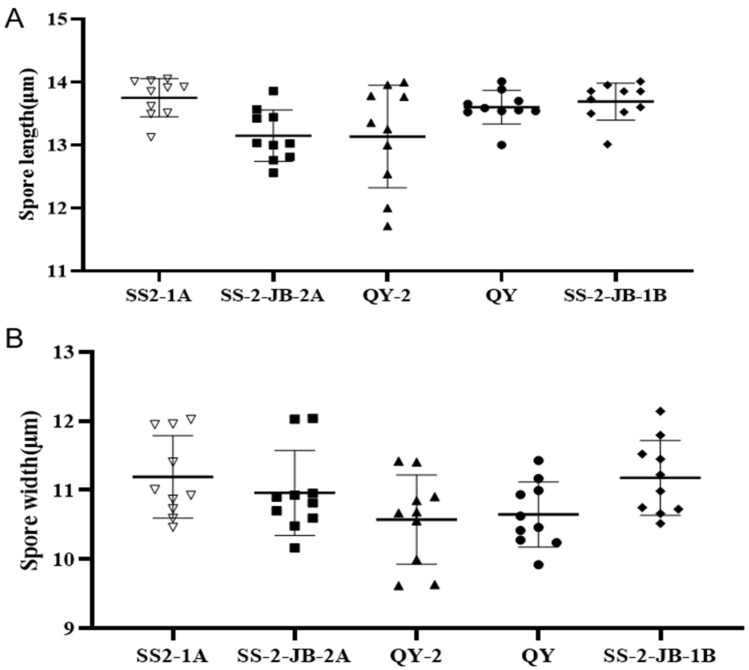
Comparison of conidia size between five *Nigrospora* isolates. The length (**A**) and the width (**B**) of the conidia were measured with the scale using a microscopy imaging system (Leica, Germany). The graph was drawn based on these data using Prism GraphPad 8.0 [26]. SS2-1A and SS-2-JB-2A: *N. oryzae* isolates; SS-2-JB-1B: *N. sphaerica*; QY: *N. chinensis*; QY-2: *N. anhuiensis* isolate. The shapes on the graph indicate the size of each conidia.

**Figure 5 jof-10-00156-f005:**
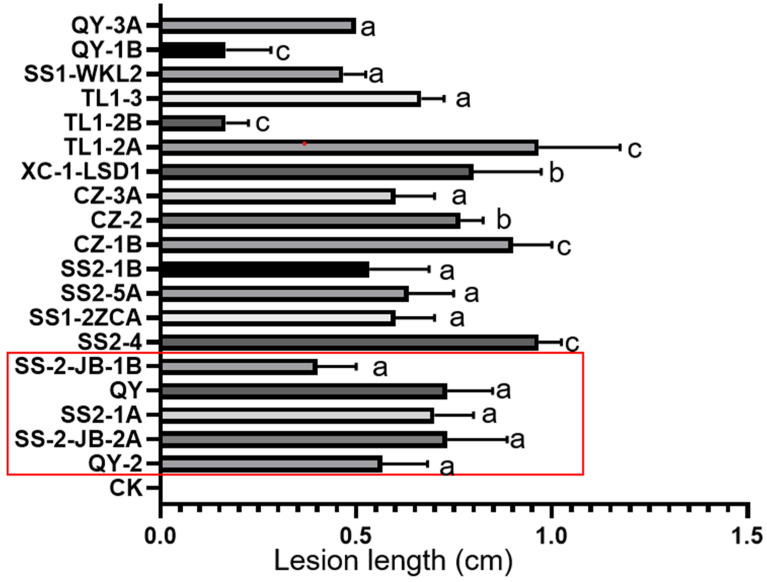
Virulence of 19 isolates on the detached rice leaves. Isolates were grown on PDA for 2 days. Then, the fungal agar plugs from the colony rim were used to inoculate the 6-week-old rice leaves, and sterile agar plugs were used as the control (CK). Lesion lengths caused by each tested isolate were measured at 2 DPI. The lesion length data were analyzed, and the graph was drawn using Prism GraphPad 8.0 [26]. One-way ANOVA analysis was performed for lesion lengths using SPSS (SPSS version 25; IBM, Chicago, IL, USA) and QY-2 as the reference isolate. The red box indicates the five isolates from different groups that were also used for morphological analysis. a indicates *p* > 0.05, b indicates *p* < 0.05, and c means *p* < 0.01.

**Figure 6 jof-10-00156-f006:**
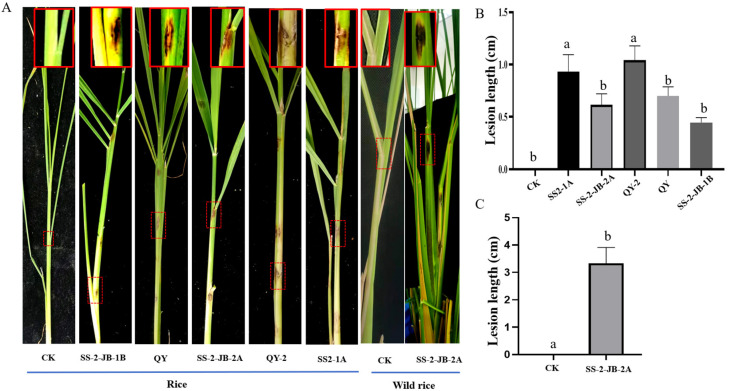
In vivo virulence of the novel *N. anhuiensis* QY-2 and other isolates. (**A**) The symptoms caused by five isolates on rice and wild rice. The fungal agar plugs were inoculated on the straw of the six-week-old rice leaves and three-month-old wild rice shoots, and sterile agar plugs were used as the control (CK). Lesion sizes were measured, and pictures were taken at 7 days post-inoculation. (**B**) The difference in lesion sizes between different isolates. The lesion length was analyzed and imaged using Prism GraphPad 8.0 [26]. QY-2 is the reference isolate. (**C**) Lesion length caused by *N. oryzae* isolate SS-2-B-2A. The fungal agar plugs were used to inoculate the straw of three-month-old wild rice shoots, and sterile agar plugs were used as the control. a indicates *p* > 0.05, b indicates *p* < 0.01.

**Figure 7 jof-10-00156-f007:**
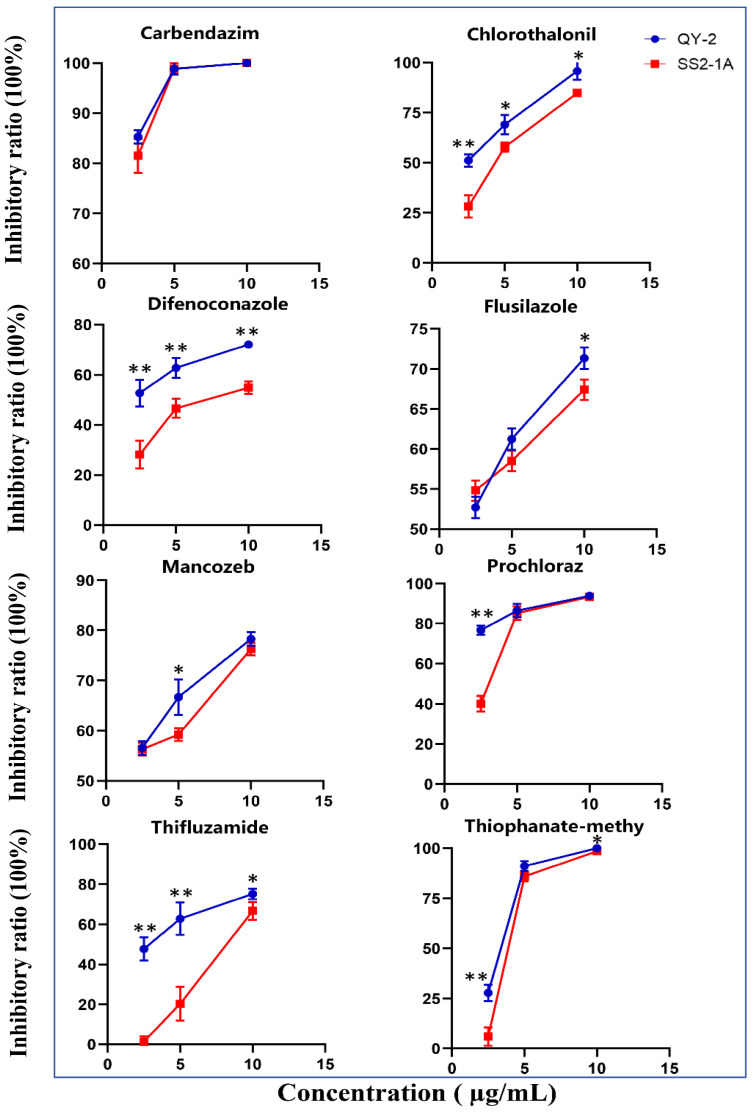
Effects of different fungicides on *N. anhuiensis* isolate QY-2. The *N. oyrzae* (as the control) isolate SS2-1A and *N. anhuiensis* isolate QY-2 were cultured on PDA, adding different fungicides (carbendazim, mancozeb, difenoconazole, chlorothalonil, thiophanate–methyl, flusilazole, prochloraz, and thifluzamide) for 2 days at 25 °C. The colony diameter was recorded, and the data were visualized using GraphPad Prism 8.0 software [26]. One-way ANOVA analysis was performed for each concentration of the fungicides using SPSS (SPSS version 25; IBM, Chicago, IL, USA), and SS2-1A was used as the reference isolate. * indicates *p* < 0.05 and ** indicates *p* < 0.01.

**Table 1 jof-10-00156-t001:** Accession numbers of genes that used for phylogenetic analysis.

Species	Isolate ID	GenBank Accession Numbers		
		ITS	TUB2	TEF1-α
*N. oryzae*	AQ-2	OP677959	PP103591	PP103566
	CZ-1A	OP677960	PP103592	PP103567
	CZ-1B	OP677961	PP103593	PP103568
	CZ-3A	OP677963	PP103594	PP103569
	CZ-3B	OP677964	PP103595	PP103570
	QY-1A	OP677967	PP103596	PP103571
	QY-1B	OP677968	PP103597	PP103572
	QY-3A	OP677970	PP103598	PP103573
	QYJ-4	OP677971	PP103599	PP103574
	SS1-2ZCA	OP677972	PP103600	PP103575
	SS1-2ZCB	OP677973	PP103601	PP103576
	SS1-WKL2	OP677974	PP112992	PP103577
	SS2-1A	OP677975	PP103602	PP103578
	SS2-1B	OP677976	PP103603	PP103579
	SS2-4	OP677977	PP103604	PP103580
	SS2-5A	OP677978	PP103605	PP103581
	SS-2-JB-2A	OP677981	PP103606	PP103582
	TL1-2A	OP677982	PP103607	PP103583
	TL1-2B	OP677983	PP103608	PP103584
	TL1-3	OP677984	PP103609	PP103585
	XC-1-LSD1	OP677985	PP103610	PP103586
	XC-LSD2	OP677986	PP103611	PP103587
*N. chinensis*	QY	OP677966	PP103612	PP103588
*N. sphaerica*	SS-2-JB-1B	OP677980	PP103613	PP103589
*Nigrospora* sp.	QY-2	OP677969	PP103614	PP103590

Note: the rest of genes were downloaded from NCBI according to previous study [23]. The new isolate QY-2 was recorded as *Nigrospora* sp. in NCBI gene bank, but when the paper is published, the administrator will change the name.

**Table 2 jof-10-00156-t002:** Origin of newly sampled *Nigrospora* isolates.

ID	Date	Collection Area	Latitude and Longitude	Host
SS1-2ZCA	2021.09	Yijiang District, Wuhu City *	118°19′20.39″ E, 31°9′27.88″ N	Grass
SS1-2ZCB	2021.09	Yijiang District, Wuhu City *	118°19′20.39″ E, 31°9′27.88″ N	Grass
SS1-WKL2	2021.06	Yijiang District, Wuhu City *	118°19′20.39″ E, 31°9′27.88″ N	Rice
SS2-1A	2021.09	Yijiang District, Wuhu City *	118°20′8.65″ E, 31°6′51.92″ N	Rice
SS2-1B	2021.09	Yijiang District, Wuhu City *	118°20′8.65″ E, 31°6′51.92″ N	Rice
SS2-4	2021.09	Yijiang District, Wuhu City *	118°20′8.65″ E, 31°6′51.92″ N	Rice
SS2-5A	2021.09	Yijiang District, Wuhu City *	118°20′8.65″ E, 31°6′51.92″ N	Rice
SS-2-JB-1B	2021.09	Yijiang District, Wuhu City *	118°20′8.65″ E, 31°6′51.92″ N	Wild rice
SS-2-JB-2A	2021.09	Yijiang District, Wuhu City *	118°20′8.65″ E, 31°6′51.92″ N	Wild rice
TL1-2A	2021.10	Jiao District, Tongling City *	117°37′29.60″ E, 30°48′49.95″ N	Rice
TL1-2B	2021.10	Jiao District, Tongling City *	117°37′29.60″ E, 30°48′49.95″ N	Rice
TL1-3	2021.10	Jiao District, Tongling City *	117°37′29.60″ E, 30°48′49.95″ N	Rice
CZ-1A	2021.10	Guichi District, Chizhou City *	117°26′10.27″ E, 30°38′51.64″ N	Rice
CZ-1B	2021.10	Guichi District, Chizhou City *	117°26′10.27″ E, 30°38′51.64″ N	Rice
CZ-3A	2021.10	Guichi District, Chizhou City *	117°26′10.27″ E, 30°38′51.64″ N	Rice
CZ-3B	2021.10	Guichi District, Chizhou City *	117°26′10.27″ E, 30°38′51.64″ N	Rice
QY-1A	2021.10	Qingyang County, Chizhou City *	117°55′15.66″ E, 30°34′58.69″ N	Rice
QY-1B	2021.10	Qingyang County, Chizhou City *	117°55′15.66″ E, 30°34′58.69″ N	Rice
QY-3A	2021.10	Qingyang County, Chizhou City *	117°55′15.66″ E, 30°34′58.69″ N	Rice
QY-1	2021.07	Qingyang County, Chizhou City *	117°53′57.06″ E, 30°36′45.36″ N	Rice
QY-2	2021.07	Qingyang County, Chizhou City *	117°54′49.77″ E, 30°36′5.07″ N	Rice
XC-1-ZSD1	2021.06	Xuanzhou District, Xuancheng City *	118°36′24.74″ E, 30°52′33.83″ N	Rice
XC-ZSD2	2021.06	Xuanzhou District, Xuancheng City *	118°36′24.74″ E, 30°52′33.83″ N	Rice
AQ-2	2021.06	Huaining County, Anqing City *	116°41′18.26″ E, 30°27′50.35″ N	Rice
QYJ-4	2021.06	Huaining County, Anqing City *	116°24′15.90″ E, 30°27′50.35″ N	Rice

* Anhui province of China.

## Data Availability

All the data are covered in the paper and graphs.

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
