# Peer review of "Identification and Characterization of Nigrospora Species and a Novel Species, Nigrospora anhuiensis, Causing Black Leaf Spot on Rice and Wild Rice in the Anhui Province of China"

_jof, 2024, doi:10.3390/jof10020156_

Round 1

Reviewer 1 Report

Comments and Suggestions for Authors

General comments

The manuscript presents an investigation of Nigrospora spp. diversity on rice and wild rice in the Anhui province of China. The study includes isolation and characterization of 25 isolates. The species identification is base on multigene sequencing and morphological characterization. N. oryzae was determined as the most abundant species in the investigated area according to the study. In addition to known members of the genus Nigrospora, N. oryzae, N. shaerica and N. chinensis, a novel species N. anhuiensis was also identified. It was differentiated morphologically from other isolated species by formation of dark green colony center after four days incubation on PDA, as well as on the base of phylogenetic analysis. Pathogenicity of selected isolates representing the studied fungal species was proved by detached leaves and seedlings of rice and wild rice. Sensitivity of the novel species N. anhuiensis to eight different fungicides was also tested and carbendazim was determined as the most effective for the pathogen control.

The manuscript is well constructed. The topic corresponds to the research scope of the Journal of Fungi.

Specific comments

Abstract and Introduction sections are informative and clearly presented. The Material and Methods are described in detail with one exception concerning the testing of virulence that needs of some additional information about control plants that were used and how old were the plants at the time of inoculation (see below). The results are presented in correct order. The Figures are representative and visualized the results clearly, but their descriptions need of corrections and Figure 6 should be specified (see below). The Discussion more or less repeats the data from the other parts of the manuscript and should be improved in the revised version to point out benefits of the study on the background of other investigations on Nigrospora spp. Linkage to former papers characterizing N. sphaerica and N. chinensis that are reported in the manuscript for the first time as a rice pathogen will be useful too. Conclusions summarized the most important data of the research. References are up to date.

Some recommendations to the authors and inaccuracies are listed below.

Page 5, Row 151

Fig. 1 needs of more precise title. Letters A, B and C should be added on photos.

Page 7, Rows 205-211

Fig. 3 - incorrect title and description. The text is the same as the one for the Fir. 2.

Page 8, Row 213

Fig. 4 represents not only conidia size of the novel N. anhuiensis QY-2, but also other tested species. Title should be specified in the revise version of the manuscript.

Page 8, Rows 217-224

There is no information about controls used in the virulence testing.

Page 9, Rows 227-228

The description of Fig. 5 is that leaves of the 6-week-old rice were inoculated with the tested fungi, however according to the Material and Methods section (page 3, rows 98-99) one-month-old rice seedlings were used for detached leaves and in vivo experiments in the virulence testing. The information about inoculation procedure should be presented correctly.

Page 10

Fig. 6 should be modified, starting with the title – it’s not only for in vivo pathogenicity tests of QY-2, but there is a data for other isolates too.

The same comments about description of the Fig. 6 (rows 244-248) as for Fig. 5.

The list tested plants on photos not correspond to the graph, where a data for SS-2-JB-2A is missing and have to be added.

My advice to the authors is to rearrange the photos of infected plants in the same order as on the graph to be easier for readers to compare the results for each isolate and the respective species.

Page 11, Rows 292-293

The sentence is not clear enough.

Page 11, Rows 301-309

This paragraph of the Discussion is not well organized and presents only the information that we already know from the Results.

Author Response

Quality of English Language

( ) I am not qualified to assess the quality of English in this paper
( ) English very difficult to understand/incomprehensible
( ) Extensive editing of English language required
( ) Moderate editing of English language required
( ) Minor editing of English language required
(x) English language fine. No issues detected

Yes

Can be improved

Must be improved

Not applicable

Does the introduction provide sufficient background and include all relevant references?

(x)

( )

( )

( )

Are all the cited references relevant to the research?

(x)

( )

( )

( )

Is the research design appropriate?

(x)

( )

( )

( )

Are the methods adequately described?

( )

(x)

( )

( )

Are the results clearly presented?

( )

(x)

( )

( )

Are the conclusions supported by the results?

(x)

( )

( )

( )

Comments and Suggestions for Authors

Response: Thanks for your positive comments and many good suggestions to our manuscript, which really help us a lot to improve the quality of the paper. We already revised the paper, please check. 

General comments

The manuscript presents an investigation of Nigrospora spp. diversity on rice and wild rice in the Anhui province of China. The study includes isolation and characterization of 25 isolates. The species identification is base on multigene sequencing and morphological characterization. N. oryzae was determined as the most abundant species in the investigated area according to the study. In addition to known members of the genus NigrosporaN. oryzaeN. shaerica and N. chinensis, a novel species N. anhuiensis was also identified. It was differentiated morphologically from other isolated species by formation of dark green colony center after four days incubation on PDA, as well as on the base of phylogenetic analysis. Pathogenicity of selected isolates representing the studied fungal species was proved by detached leaves and seedlings of rice and wild rice. Sensitivity of the novel species N. anhuiensis to eight different fungicides was also tested and carbendazim was determined as the most effective for the pathogen control.

The manuscript is well constructed. The topic corresponds to the research scope of the Journal of Fungi.

Specific comments

Abstract and Introduction sections are informative and clearly presented. The Material and Methods are described in detail with one exception concerning the testing of virulence that needs of some additional information about control plants that were used and how old were the plants at the time of inoculation (see below). The results are presented in correct order. The Figures are representative and visualized the results clearly, but their descriptions need of corrections and Figure 6 should be specified (see below). The Discussion more or less repeats the data from the other parts of the manuscript and should be improved in the revised version to point out benefits of the study on the background of other investigations on Nigrospora spp. Linkage to former papers characterizing N. sphaerica and N. chinensis that are reported in the manuscript for the first time as a rice pathogen will be useful too. Conclusions summarized the most important data of the research. References are up to date.

Response: Yes, like you said, this is an interesting work. We are sorry about the errors or confusion in the form version. Now, we already revised the manuscript according to your suggestions.

Some recommendations to the authors and inaccuracies are listed below.

Page 5, Row 151

Fig. 1 needs of more precise title. Letters A, B and C should be added on photos.

Response: We already revised the title and added the graph note A,B,C on the photo.

 Page 7, Rows 205-211

Fig. 3 - incorrect title and description. The text is the same as the one for the Fir. 2.

Response: I am sorry about the mistakes, we already modified.

 Page 8, Row 213

Fig. 4 represents not only conidia size of the novel N. anhuiensis QY-2, but also other tested species. Title should be specified in the revise version of the manuscript.

Response: We have changed the title, please check.

 Page 8, Rows 217-224

There is no information about controls used in the virulence testing.

Response: We have done the virulence of the control, there were no lesion observed. Initially, we think the data were not necessary, now we have added the control, and added the description for the control. 

 Page 9, Rows 227-228

The description of Fig. 5 is that leaves of the 6-week-old rice were inoculated with the tested fungi, however according to the Material and Methods section (page 3, rows 98-99) one-month-old rice seedlings were used for detached leaves and in vivo experiments in the virulence testing. The information about inoculation procedure should be presented correctly.

 Response: Sorry about the confusions, we used the 6-week-old rice plants, and now we have unified seedling’s age.

Page 10

Fig. 6 should be modified, starting with the title – it’s not only for in vivo pathogenicity tests of QY-2, but there is a data for other isolates too.

Response: We have revised the title, please check in the text.

The same comments about description of the Fig. 6 (rows 244-248) as for Fig. 5.

Response: We have changed the title according to your suggestions.

The list tested plants on photos not correspond to the graph, where a data for SS-2-JB-2A is missing and have to be added.

Response: We already added the photos for wild rice and the graph were also changed according to your suggestions.

My advice to the authors is to rearrange the photos of infected plants in the same order as on the graph to be easier for readers to compare the results for each isolate and the respective species.

Response: Yes, you are right, rearrangement of the photos should be beneficial for easier understanding. We already rearranged the stains order, and marked in the gragh.

 Page 11, Rows 292-293

The sentence is not clear enough.

Response: we have revised, please check.

 Page 11, Rows 301-309

This paragraph of the Discussion is not well organized and presents only the information that we already know from the Results.

Response: I did some revision about the discussion, I hope now it is good.

Reviewer 2 Report

Comments and Suggestions for Authors

This is an interesting study. This manuscript “Identification and characterization of Nigrospora spp. and a novel species Nigrospora wuhuensis causing black leaf spot on rice and wild-rice in the Anhui province of China” provides a new species of Nigrospora using morphological and molecular identification. Overall, the manuscript present considerable results but unfortunately is poorly presented in a way that cannot be easily understood. For example:

In the title the authors mentioned Nigrospora wuhuensis then in the text we cannot found it and we can find the new species is Nigrospora anhuiensis, how the reader can understand which is the new species if the title didn’t explain the manuscript!

Results should be more clear and more detailed, morphological identification: how much conidia you measured to get the size of conidia? You should mention in the material and methods. scale bar it is very important in your figure for morphological identification

All the sequencing needs to be submitted in NCBI National Center for Biotechnology Information before publication otherwise you cannot consider new species,

Add all accession number to the manuscript

The analysis statistic it is very important please include in you manuscript

Author Response

Quality of English Language

(x) I am not qualified to assess the quality of English in this paper
( ) English very difficult to understand/incomprehensible
( ) Extensive editing of English language required
( ) Moderate editing of English language required
( ) Minor editing of English language required
( ) English language fine. No issues detected

Yes

Can be improved

Must be improved

Not applicable

Does the introduction provide sufficient background and include all relevant references?

( )

( )

(x)

( )

Are all the cited references relevant to the research?

( )

( )

(x)

( )

Is the research design appropriate?

( )

( )

(x)

( )

Are the methods adequately described?

( )

( )

(x)

( )

Are the results clearly presented?

( )

( )

(x)

( )

Are the conclusions supported by the results?

( )

( )

(x)

( )

Comments and Suggestions for Authors

This is an interesting study. This manuscript “Identification and characterization of Nigrospora spp. and a novel species Nigrospora wuhuensis causing black leaf spot on rice and wild-rice in the Anhui province of China” provides a new species of Nigrospora using morphological and molecular identification. Overall, the manuscript present considerable results but unfortunately is poorly presented in a way that cannot be easily understood. For example:

Response: We agree with your points and thanks for your suggestions. The previous version of the paper was not good enough. We have done a lot of revisions, please check. We hope now it meets the standard for publish.

In the title the authors mentioned Nigrospora wuhuensis then in the text we cannot found it and we can find the new species is Nigrospora anhuiensis, how the reader can understand which is the new species if the title didn’t explain the manuscript!

Response: We are sorry about the mistakes, actually, at the beginning, we named it as Nigrospora wuhuensis, but one author said it is not found in Wuhu, that is not accurate. So we changed the name to Nigrospora anhuiensis, but we forgot to change the title. Now, we already changed the title, and we make sure the current name is correct.

Results should be more clear and more detailed, morphological identification: how much conidia you measured to get the size of conidia? You should mention in the material and methods. scale bar it is very important in your figure for morphological identification

Response: Yes, you are right. We have already added the information for conidia and the scale bar for each photo were also added.

All the sequencing needs to be submitted in NCBI National Center for Biotechnology Information before publication otherwise you cannot consider new species,

Add all accession number to the manuscript

Response: Yes, we agree with you. We have already added a table (Table 1) for the accession numbers. All the sequences we generated are presented in the table, please check.

The analysis statistic it is very important please include in you manuscript

Response: we have reanalyzed the data, and the analysis results were already shown on the new photos.

Reviewer 3 Report

Comments and Suggestions for Authors

The manuscript reports on research investigating Nigrospora species from rice plants in Anhui province. Three previously observed species were isolated and a new species was identified.  There are some strong research aspects to the paper, such as the comparisons of the new species with others, pathogenicity testing, measured of fungicide responses and others.  However, there are two major issues.

(1)   The new species has not been formally named.  This requires registration of the name at somewhere like MycoBank, depositing a type specimen to a herbarium (or equivalent) and writing a diagnosis/description of the species (some of that text is there).  There are plenty of examples of how this is styled.

(2)   The text requires extensive editing, as numerous typographical errors.  Perhaps the most obvious is the species is called N. wuhuensis in the title!.  A few other ones are in the next section.

Comments on the Quality of English Language

Line 30: ‘more and more famous’ is odd language.

Line 94: the NCBI accessions should be provided.

Figure 2 seems low resolution.

Line 176: ‘Size marker shows.’ is missing text.

Line 204: title is for figure 2, as this is morphology.

Line 213: Title is not accurate, more like ‘Comparison of conidia size between five strains of Nigrospora’.

Line 99 is one month and lines 228 and 245 is 6 weeks for plant age.

Figure 7 is small, and could be redesigned to be larger and therefore clear in print.

Line 272: add italics to the species names.

Line 306: Koch’s.

Line 329: ‘cities’.

Citation list will need editing as well, e.g. adding italics to species names, correct use of capital letters.

Reference 8: ‘First report of Davidia involucrata leaf blight caused by Nigrospora oryzae in Sichuan, China’

Reference 20: missing the species name.

Author Response

Quality of English Language

( ) I am not qualified to assess the quality of English in this paper
( ) English very difficult to understand/incomprehensible
( ) Extensive editing of English language required
(x) Moderate editing of English language required
( ) Minor editing of English language required
( ) English language fine. No issues detected

Yes

Can be improved

Must be improved

Not applicable

Does the introduction provide sufficient background and include all relevant references?

(x)

( )

( )

( )

Are all the cited references relevant to the research?

(x)

( )

( )

( )

Is the research design appropriate?

(x)

( )

( )

( )

Are the methods adequately described?

(x)

( )

( )

( )

Are the results clearly presented?

( )

(x)

( )

( )

Are the conclusions supported by the results?

( )

( )

(x)

( )

Comments and Suggestions for Authors

The manuscript reports on research investigating Nigrospora species from rice plants in Anhui province. Three previously observed species were isolated and a new species was identified.  There are some strong research aspects to the paper, such as the comparisons of the new species with others, pathogenicity testing, measured of fungicide responses and others.  However, there are two major issues.

Response: many thanks for your positive comments and the suggestions. 

(1)   The new species has not been formally named.  This requires registration of the name at somewhere like MycoBank, depositing a type specimen to a herbarium (or equivalent) and writing a diagnosis/description of the species (some of that text is there).  There are plenty of examples of how this is styled.

Response: we have submitted the detailed description of the new species to fungal names, and the strain registration was also added in the text.

(2)   The text requires extensive editing, as numerous typographical errors.  Perhaps the most obvious is the species is called N. wuhuensis in the title!.  A few other ones are in the next section.

Response: We are sorry about the mistakes. The senior professor Jan Schirawski has edited the whole paper, we hope the current version is good now. For the name, actually, at the beginning, we named it as Nigrospora wuhuensis, but one author said it is not found in Wuhu, that is not accurate. So we changed the name to Nigrospora anhuiensis, but we forgot to change the title. Now, we already changed the title, and we make sure the current name is correct.

Comments on the Quality of English Language

Line 30: ‘more and more famous’ is odd language.

Response: We deleted these words “more and more”

Line 94: the NCBI accessions should be provided.

Response: Yes, you are right, we have added the NCBI accessions in Table 1.

Figure 2 seems low resolution.

Response: Yes, the picture is not very clear, we changed this pictures using the original picture with high resolution.

Line 176: ‘Size marker shows.’ is missing text.

Response: We have added the missing text.

Line 204: title is for figure 2, as this is morphology.

Response: I guess you said the Figure 3, the title and text was wrong, We already revised.

Line 213: Title is not accurate, more like ‘Comparison of conidia size between five strains of Nigrospora’.

Response: we have revised the title according to your suggestions.

Line 99 is one month and lines 228 and 245 is 6 weeks for plant age.

Response: we sorry about the confusion, actually is 6 weeks, we already revised in the method.

Figure 7 is small, and could be redesigned to be larger and therefore clear in print.

Response: Yes, it looks like a little small, we already changed the photos, and we hope it is better.

Line 272: add italics to the species names.

Response: we already modified these species names according to your suggestions.

Line 306: Koch’s.

Response: we already revised according to your suggestions.

Line 329: ‘cities’.

Response: we already corrected the wrong word.

Citation list will need editing as well, e.g. adding italics to species names, correct use of capital letters.

Reference 8: ‘First report of Davidia involucrata leaf blight caused by Nigrospora oryzae in Sichuan, China’

Reference 20: missing the species name.

Response: we have revised according to your suggestions.

Reviewer 4 Report

Comments and Suggestions for Authors

The work expands knowledge about the variability of the genus Nigrospora, but is very poorly prepared

3 „novel species Nigrospora wuhuensis”  - wrong species name

15 „Nigrospora shaerica”  - wrong species name

32 – 37  uniformity in host names should be maintained, English or Latin names should be used

92  “The phylogenetic tree was generated using the neighborjoining method with 1000 bootstrap replicates with the MEGA7.0 software and TUB2 sequences of the new N. anhuiensis isolate were submitted to the NCBI database.”

GenBank accessions of the generated sequences of all genes must be added in separated table

190 “and septate hyphae (Figure 3B)”

the quality of the photos does not allow to determine whether the hyphae are septated

191 “Conidia of all investigated strains had a similar globose or oval, black, smooth and aseptate morphology (Figure 3C)”

the quality of the photos does not allow for the characterization of the spores

204 “Figure 3 Multigene phylogenetic analysis for species identification”.

Figugre doesn’t show multigene phylogenetic analysis 

231 Fig 4

instead of isolate numbers, the species names should be given

Results regarding virulence and susceptibility to fungicides is not supported by statistics. There is no information whether the differences found were statistically significant

The work does not meet the standards related to the description of a new species, which should include: authors of a new taxon, etymology of the new name, indication of the type isolate and where it was deposited, precise characterization of macro and microscopic features, indications of differences from related species

The study which  may serve as an example of how to describe a new species is the work of Wang, M.; Liu, F.; Crous, P.W.; Cai, L. Phylogenetic reassessment of nigrospora: Ubiquitous endophytes, plant and human pathogens. Persoonia-Molecular Phylogeny and Evolution of Fungi 2017, 39, 118-142.

Author Response

Open Review

Quality of English Language

(x) I am not qualified to assess the quality of English in this paper
( ) English very difficult to understand/incomprehensible
( ) Extensive editing of English language required
( ) Moderate editing of English language required
( ) Minor editing of English language required
( ) English language fine. No issues detected

Yes

Can be improved

Must be improved

Not applicable

Does the introduction provide sufficient background and include all relevant references?

(x)

( )

( )

( )

Are all the cited references relevant to the research?

(x)

( )

( )

( )

Is the research design appropriate?

( )

( )

(x)

( )

Are the methods adequately described?

( )

(x)

( )

( )

Are the results clearly presented?

( )

( )

(x)

( )

Are the conclusions supported by the results?

( )

(x)

( )

( )

Comments and Suggestions for Authors

The work expands knowledge about the variability of the genus Nigrospora, but is very poorly prepared

Response: many thanks for your time and the suggestions to our manuscript. 

3 „novel species Nigrospora wuhuensis”  - wrong species name

15 „Nigrospora shaerica”  - wrong species name

Response: we have modified these wrong species name.

32 – 37  uniformity in host names should be maintained, English or Latin names should be used

Response: we have revised this part according to your suggestions.

92  “The phylogenetic tree was generated using the neighborjoining method with 1000 bootstrap replicates with the MEGA7.0 software and TUB2 sequences of the new N. anhuiensis isolate were submitted to the NCBI database.”

GenBank accessions of the generated sequences of all genes must be added in separated table

Response: Yes, we agree with you, the GenBank accessions were already added in Table 1, please check.

190 “and septate hyphae (Figure 3B)”

the quality of the photos does not allow to determine whether the hyphae are septated

Response: we have added the photos for hyphal septate through a small picture.

191 “Conidia of all investigated strains had a similar globose or oval, black, smooth and aseptate morphology (Figure 3C)”

the quality of the photos does not allow for the characterization of the spores

Response: we changed some of the photos, and re-edited the photos, I hope the current graph is clear.

204 “Figure 3 Multigene phylogenetic analysis for species identification”.

Figugre doesn’t show multigene phylogenetic analysis 

Response: You are right, we made a mistake, actually, we used a multi-locus combined sequence for constructing a tree. We already revised the text.

231 Fig 4 instead of isolate numbers, the species names should be given

Response: Since the previous graph we used showed the isolate numbers, for uniformity, we think the picture should not be changed, but we added the strains’ species names in the graph note.

Results regarding virulence and susceptibility to fungicides is not supported by statistics. There is no information whether the differences found were statistically significant

Response: Yes, you are right, we already added the statistically analysis, please check the new version.

The work does not meet the standards related to the description of a new species, which should include: authors of a new taxon, etymology of the new name, indication of the type isolate and where it was deposited, precise characterization of macro and microscopic features, indications of differences from related species

The study which  may serve as an example of how to describe a new species is the work of Wang, M.; Liu, F.; Crous, P.W.; Cai, L. Phylogenetic reassessment of nigrospora: Ubiquitous endophytes, plant and human pathogens. Persoonia-Molecular Phylogeny and Evolution of Fungi 2017, 39, 118-142.

Response: we descripted some features for the new species in the discussion part, and we also registered the new species on Fungal Names, where the audience could find the details for the new species.

Round 2

Reviewer 2 Report

The manuscript is improved after revision but some points need to be clarified

In table 1 the GenBank accession numbers OP677969 PP103614 PP103590 is submitted as Nigrospora sp. Please mention the same in the table and not N. anhuiensis , the caption of the table need to be more developed,

Explain in the Materials and methods section the analysis statistics

Figure 3 please mention the size of each scale bar

Figure5 please develop more the caption and mention the test used to compare data and the same for Figure 

Author Response

Open Review

 I would not like to sign my review report

 I would like to sign my review report

Does the title describe the article's topic with sufficient precision?

Yes

No

Does the introduction provide a comprehensive yet concise overview about the state of knowledge in the area of research?

Yes

No

Is the research design appropriate and are the methods adequately described?

Yes

No

Are the results presented clearly and in sufficient detail, are the conclusions supported by the results and are they put into context within the existing literature?

Yes

No

Are all of the cited references relevant to the research?

Yes

No

Does this article provide a relevant contribution to the scientific discussion of this topic?

Yes

No

English language and style

I am not qualified to assess the quality of English in this paper

English very difficult to understand/incomprehensible

Extensive editing of English language required

Moderate editing of English language required

Minor editing of English language required

English language fine. No issues detected

Comments for Authors

Advice for completing your review can be found at: https://www.mdpi.com/reviewers#Review_Report

Major comments

The manuscript is improved after revision but some points need to be clarified

Re: Thanks for your positive comments to our first round revision.

Detail comments

In table 1 the GenBank accession numbers OP677969 PP103614 PP103590 is submitted as Nigrospora sp. Please mention the same in the table and not N. anhuiensis , the caption of the table need to be more developed,

Re: We have revised the table and the title according to your suggestions.

Explain in the Materials and methods section the analysis statistics

Re: We have added the details for statistics in the materials and methods.

Figure 3 please mention the size of each scale bar

Re: The size of each scale bar were added and even mentioned in the caption

Figure5 please develop more the caption and mention the test used to compare data and the same for Figure 

Re: We already revised the figure 5, but we are not sure what should be the best, because there are a lot of information about the test, if there are still some problems, please do not hesitate to give the comments.

Reviewer 3 Report

The manuscript has been improved since the first submission.  However, it would still benefit from another round of editing of the text, as there are numerous typographical errors still.

A few obvious ones:

L36: 'grass,cotton'

L64: 'rice[11]'

L113-114: 'potatodex-trose medium(PDA)'

L178: 'ClastalW2'

L317: 'disease[11]'

L344: 'HE'

L350: 'On CA, the conidia will produce in a week'

L364: 'all most'

L389: 'brachiaria griseb' (final version is only 'Brachiaria')

Figure 7 does not look very different, may be a bit larger but not into the margin.  A better arrangement would be rather than two rows with four result in each, four rows with two result each.  That would enable the data to be seen more clearly.

Author Response

Open Review

 I would not like to sign my review report

 I would like to sign my review report

Does the title describe the article's topic with sufficient precision?

Yes

No

Does the introduction provide a comprehensive yet concise overview about the state of knowledge in the area of research?

Yes

No

Is the research design appropriate and are the methods adequately described?

Yes

No

Are the results presented clearly and in sufficient detail, are the conclusions supported by the results and are they put into context within the existing literature?

Yes

No

Are all of the cited references relevant to the research?

Yes

No

Does this article provide a relevant contribution to the scientific discussion of this topic?

Yes

No

English language and style

I am not qualified to assess the quality of English in this paper

English very difficult to understand/incomprehensible

Extensive editing of English language required

Moderate editing of English language required

Minor editing of English language required

English language fine. No issues detected

Comments for Authors

Advice for completing your review can be found at: https://www.mdpi.com/reviewers#Review_Report

Major comments

The manuscript has been improved since the first submission.  However, it would still benefit from another round of editing of the text, as there are numerous typographical errors still.

Re: Many thanks for your suggestions and comments, which improved our manuscript a lot. And we already revised the manuscript according to your suggestion. Please see below.

Detail comments

A few obvious ones:

L36: 'grass,cotton'

Re: We added a space at this site.

L64: 'rice[11]'

Re: We have revised this citation.

L113-114: 'potatodex-trose medium(PDA)'

Re: We modified this wrong words.

L178: 'ClastalW2'

Re: We changed the word 'ClastalW2' to 'ClustalW2'.

L317: 'disease[11]'

Re: We have revised this citation.

L344: 'HE'

Re: We have revised ‘HE’ to He.

L350: 'On CA, the conidia will produce in a week'

Re: We already changed this description to ‘On the carrot medium’

L364: 'all most'

Re: We changed ‘all most’ to ‘most of’

L389: 'brachiaria griseb' (final version is only 'Brachiaria')

Re: We have corrected the reference according to your suggestion.

Figure 7 does not look very different, may be a bit larger but not into the margin.  A better arrangement would be rather than two rows with four result in each, four rows with two result each.  That would enable the data to be seen more clearly.

Re: We have re-arranged the graph according to your suggestion. We think now it should be good.